# A Qualitative Photo Elicitation Research Study to elicit the perception of young children with Developmental Disabilities such as ADHD and/or DCD and/or ASD on their participation

Marieke Coussens[1]*, Birger Destoop[1], Stijn De Baets[1], Annemie Desoete[2], Ann Oostra[3], Guy Vanderstraeten[1], Hilde Van Waelvelde[1], Dominique Van de Velde[1]

1 Department of Rehabilitation Sciences, Ghent University, Ghent, Belgium, 2 Department of Experimental Clinical and Health Psychology, Ghent University, Ghent, Belgium, 3 Department of Paediatrics and Medical Genetics, Ghent University, Ghent, Belgium

* Marieke.coussens@ugent.be

## Abstract

Participation, defined as 'involvement in life situations' according to the World Health Organisation, is a well-recognized concept and critical indicator of quality of life. In addition it has become an important outcome measure in child rehabilitation. However, little is known about the level of participation of young children with Developmental Disabilities. The aim of this study was to capture their subjective experiences of participation. An adapted informed consent based on a comic strip was used to get the children's assent. A Photo Elicitation study was used, in which photographs were taken by the children when they were involved in meaningful activities. The photographs were then used to facilitate communication with the children and to initiate in depth-interviews. Forty-seven interviews with 16 children between five and nine years were conducted based on their photographs. This method generated rich data, confirming that young children with Developmental Disabilities were able to inform us accurately on their experiences of participation. Data was analysed by means of an inductive thematic analysis. Results showed that children perceived their participation as satisfying when they can play, learn and join in family gatherings resulting in feelings of inclusion, recognition and belonging. When there are—on occasions—moments that their participation was obstructed, the children used two strategies to resolve it. Or they walked away from it and choose not to participate, or when autonomously motivated for the activity, they relied primarily on their context (i.e. mothers) as enabling their participation. Related to the data, children discussed themes related to their person, activities, connections and mediators between those themes. These themes fit well within earlier and current research on the subject of participation.

**Data Availability Statement:** All relevant data are with the manuscript and its supporting information files.

**Funding:** The authors received no specific funding for this work.

**Competing interests:** The authors have declared that no competing interests exist.

# Background

The International Classification of Functioning (ICF) represents the most up-to-date international disability framework [1, 2]. This framework resulted from a paradigm shift from a medical or genetic and causal approach (e.g. autism) to a focus on consequences for 'participation' e.g. difficulty with making friends [3–8]. According to the WHO [2], participation is defined as "involvement in a life situation".

In the last decade, it has become a well-recognized concept and critical indicator of quality of life [2] affecting the outcome of rehabilitation services [9]. The WHO, by developing the ICF, has stressed the importance of the environment in supporting or hindering participation [10]. A person's ability to participate can be seen as the result of interactions between aspects of health condition and contextual environmental and personal factors (i.e., age, sex, and motivation) [2]. King and colleagues [11] documented that participation in meaningful activities in many environments (home, school and community) was vital for children's well-being. By participating in such activities, children developed skills and capabilities, expressed creativity and enjoyment, developed their self-identity, self-esteem and emotional well-being, achieved a better mental and physical health, formed meaningful relationships and achieved a purpose and meaning in their lives [9, 12, 13].

Despite this emphasis on participation [5, 8, 14, 15] there seems to be no consensuses on how participation should be interpreted. Differences in the way participation is operationalized; (1) in objective variables (e.g. number of situations, frequency, location) or (2) in subjective variables (e.g. enjoyment, satisfaction, importance) of participation in (3) different contexts and over time [7, 16] results in study outcomes that are difficult to compare.

To date, Imms and colleagues [5] have conducted research which integrates various factors in a useful new direction. Their narrative systematic review concluded that the participation phenomenon is essentially dichotomous- requiring children to "attend" (be present) and to be "involved" (engage, experience and so forth) [15]. They further differentiated between influencing or "participation related constructs" which included preferences, sense of self and activity competence and 'participation' [5, 15]. Both Imms et al. and Maciver et al. [5, 15] highlighted the importance of a careful definition of participation. However, both researchers conducted reviews based on RCT's, intervention studies or focused solely on the school environment, leaving home and community out of the scope of research.

Another noteworthy finding in participation 'research' is that study outcome is mainly based on information 'about' and not on observations or professional practice 'with' children experiencing disabilities. This could be due to the fact that only recently, the World Health Organization (WHO) developed the International Classification of Functioning-Child Youth (ICF-CY). The development of ICF-CY originates from the need to describe the functioning of children and youth in correspondence with International Classification of Functioning (ICF). The ICF-CY offers especially more extended points of departure and greater detail for a better description of the growth and development of children and youth.

This might have resulted in the fact that up till now, the voices of children with disabilities were absent in disability studies [17]. To resolve this issue, researchers have traditionally used adults as proxies to establish the perspectives of these children [18]. Yet, research [19–21] revealed that parents were not to be considered reliable interpreters of the child's perspective.

There is increasing evidence that children with disabilities, such as Attention Deficit Hyperactivity Disorder (ADHD), Developmental Coordination Disorder (DCD) and Autism Spectrum Disorder (ASD), are limited in their participation compared to their typically developing peers [10, 22–24]. ADHD, DCD and ASD are neurodevelopmental disorders which typically onset in early childhood [25]. ADHD is defined by the presence of impairing symptoms of

inattention and/or hyperactivity–impulsivity, is present across two or more settings, and cannot be better explained by another condition [26]. ASD is characterized by enduring and impairing social communication and interaction deficits that occur across multiple contexts along with the presence of restricted, repetitive behaviors, interests or activities, or sensory symptoms [26].

DCD is characterized by marked impairment of motor coordination that significantly interferes with academic achievement, performance of activities in daily living and engagement in play [26]. It is however in young age difficult to differentiate between the three different diagnoses, and often the three coincide.

Prior to the Diagnostic and Statistical Manual for Mental Health Disorders -5[th] Edition (DSM-5) in 2013, clinicians were not allowed to make e.g. an ADHD diagnosis in the context of ASD. It was presumed that all symptoms of inattention and/or hyperactivity-impulsivity could be attributed to the ASD and not due to an additional ADHD diagnosis [27]. Similar for ADHD and DCD, resulting in comorbidity that was often unrecognized. Without this exclusion criterion, it is not surprising that currently children do get more than one diagnosis as ADHD, DCD and ASD frequently co-occur [26]. Recent studies suggest about 50% of individuals diagnosed with a DD have an additional comorbid disorder, pointing to multiple deficits and a potential link between these disorders [28, 29]. In addition, when ADHD and DCD or ASD co-occur, the outcome tends to be more severe than in isolated disorders [30], with poorer psychosocial outcomes and higher levels of depressive symptoms in children with comorbid disorders [31].

However even in isolated disabilities participation difficulties occur. Studies revealed participation problems in different social activities in older children with ADHD [32, 33], due to their poor social skills and sometimes inadequate behavior. In addition, children with DCD were found to experience difficulties with dressing up, tying shoes, catching a ball, physical education, play skills and engagement in leisure activities [34–36] due to their motor coordination problems, resulting in low participation in socially valued leisure activities, such as team sports. Various studies [37, 38] also report that individuals with ASD often have severely limited participation. They were restricted in the diversity and frequency of participation compared to typically developing peers [39, 40].

Surprisingly, very little is known on how these children with DD perceive 'participation'. The opinion of children with DD has been incorporated in the studies on the opinions of the parents and care takers, although research has shown that there might be some discordance between parents' and children's reports. Thus, to obtain the most comprehensive picture of the children's experience of participation, it is also necessary to communicate with children, to study how we can help them to reach an optimal participation across their life span [41–43]. Therefore, the main aim of this study is to investigate how young children with DD as principal stakeholders define their participation, recognizing their right to express their preferences and feelings, and to be consulted on matters that affect them [44].

## Methods

### Design

Since we were concerned with understanding the experience of participation from the children's' own frame of reference, we opted for a qualitative research design. We aimed to gain a more substantive understanding of [45–51] the unique stories of young children with DD and their views about their participation. Interviewing children can be challenging for many reasons. Difficulties in finding words, children's stranger anxiety, different use of vocabulary, lack of spontaneous conversation and distractions due to short attention span are few of them [52].

Hence, when interviewing children, different data collection strategies should be used to overcome these challenges. For this study, we specifically choose a Photo Elicitation approach.

By giving children a digital camera and by letting them collect the data, we provided prompts to assist them recall events and the use of props for non-verbal demonstration of their thoughts and events during the interview. Thus, photographs were used as a tool to elicit data, not to analyse it nor to use the photographs' as a vehicle through which findings could be represented and communicated [53] such as is expected in Participation Action Research (PAR).

The Consolidated Criteria for Reporting Qualitative Research (COREQ) were followed to guide this research [54]. Previous research has demonstrated that children as young as 4 years old can provide information regarding their daily lives and health experiences but only when the method has been adapted to their possibilities [36, 52]. Therefore, data were collected using a Photo Elicitation study and has been described already as an effective method for studying the lived experiences of children with disabilities, particularly those with communication impairments [55]. The children received a digital camera and were asked to take photos or make movies of activities they did at home, in school and in their community during a week. As most young children in Belgium are not yet owners of a smartphone, we chose to collect data with a robust digital camera that is child friendly. These photographs were used to initiate in-depth interviews [56]. More expressively efficient than mere words, the photos taken by the children were used to ease communication during the interviews [56].

During this interview, children were asked to describe their pictures and were prompted to reveal the thoughts and feelings the pictures evoked. This approach enabled a child-centered and avoided a researcher-centered perspective [57–59].

Data were transcribed verbatim and analyzed by means of an inductive thematic analysis. Although the images are central, text (e.g. interview) remained the unit of analysis. We chose for a thematic analysis as it is still the most useful in capturing the complexities of meaning within a textual data set [60].

The entire protocol exist out of 6 steps from participant recruitment to data-analysis and is described in the following paragraphs. The protocol is schematically presented in Fig 1. This research was approved by the Ethical Committee of Ghent University (B670201835100)

**Phase 1: Participants recruitment.** Children were recruited through their parents, using an indirect recruitment strategy. Parents with children with DD were indirectly searched for via referrals from rehabilitation centers, parents' associations, and family support services located in the Flemish speaking part of Belgium by means of a leaflet containing information about the study and contact-details of the first author. Aside from this indirect way of recruiting, twelve Facebook self-advocacy groups were used for direct recruitment of the parents and their children with DD using the same (but digital) leaflet. The inclusion criteria for the children were: (1) aged between 4 and 9 years of age, (2) being diagnosed in a recognized diagnostic center with ADHD or DCD or ASD and another comorbid diagnosis such as ADHD, DCD and/or ASD (3) being diagnosed with ADHD, DCD or ASD and still in the diagnostic process of another diagnose such as ASD, ADHD and/or DCD (4) having a normal Intelligence Quotient (IQ) (defined by the recognized diagnostic center) and (5) being able to verbally communicate age accordingly (defined by the parents).

**Phase 2: Informed Consent (IC) from the parent.** The parents that made themselves known were contacted by the researcher, received an explanation about the research project and an explanation on how the children will be informed about the project.

In line with Whyte [17] and Curtin [61], the parents were explained that also the children were asked to give their consent by means of an Informed Assent (IA). Therefore, it was explained to the parents that an 'easy to understand comic strip book format' (Rosenfeld et al.,

**Phase 1**
- Participants Recruitment
  - Indirect via referals
  - Direct via Facebook

**Phase 2**
- IC from Parents
  - Telephone interview with parents
  - An explanation to the parents how a comic strip is used as a way to inform the child regarding the research project
  - Collect IC from the parents

**Phase 3**
- IA from child
  - The comic strip book is read to the child by the parent during 1 week
  - The IA is obtained from the child
  - The researcher visits the child
  - The child's favorite game is played to get acquainted
  - The use of the camera is explained

**Phase 4**
- In-depth interview 2 based on photographs
- The researcher visits the child a second time
- The researcher re-explains the right to withdraw
- The interview starts based on pictures taken by the child
- A parent(s) or grandparent is nearby during the interview

**Phase 5**
- Member check and Interview 3 with presence of parent
- The researcher visits the child a third time to do a member check
- The goal is to get the child's (dis) approval and when needed, get more information from the child and the parent

**Phase 6**
- Data Analysis and Dissimination
- Data analyses
- Data disseminatie to the parents by lettre (easy to read format)
- Data disseminatie to the child in the form of drawings

**Fig 1. Six phases of the study.**

2018) will be used to explain the entire research project to the child and was asked whether they agreed with it to read it together with the child. Once the parents agreed on doing so, and had a clear view on the protocol of the research project they were asked to participate in the research. If the agreed, they were asked to sign an Informed Consent (IC).

**Phase 3: Informed Assent (IA) from the child and interview 1.** The parents were asked to carefully read the IA together with the child. In doing so, the child had time to think about the project before the first contact with the researcher. After that, the child was asked whether he or she agreed with that and the IA was obtained. To establish a relationship and to create trust, a favorite game of the child was played between the researcher and the child. The stories the child told during the game was seen as data, and consequently, this game and the conversation that went along with it was seen as the first interview. As a last step in this phase, the use of the camera was explained.

During the first interview, we explained the children the use of the camera. While doing so, we also mentioned that they needed to ask permission first of the person that they wanted to take a picture from. As well, we made information letters for the school and the sports clubs or other recreational settings so that these settings were informed and were able to consent voluntary. We explained that they could take pictures of activities and contexts in which they participated. We did this in order to help the children with recall to stimulate conversation during the follow-up interview.

**Phase 4: In-depth interview 2 based on the photographs.** Data were collected from September 2017-March 2018 by the first two authors whereby the researcher tried to gauge deeper into the thoughts, emotions and perception of the child while focusing on pictures from the camera assignment.

The child could choose which pictures he would like to talk about. That approach gave the child the opportunity to communicate and/or to elaborate on their experiences with their activities. Only when the child had difficulties in finding ways to express its thoughts and emotions, the researcher asked the (grant-) parent to help with some guiding questions or to give an answer from their perspective. We expected that the conversation between parent and child could lead to a deeper understanding of the thoughts and feelings of the child. All interviews took place in the home of the child and in the presence of the (grand-) parent to ensure a familiar and safe environment. All interviews were video-recorded to make a conversion to a transcript possible. Two researchers collected the data and followed the interview protocol (see S1 Appendix).

**Phase 5: Member check and interview 3.** In this follow-up interview, a member check took place via 1) showing the selected pictures again, and 2) giving a short overview of the transcripts of the previous interview. The children were asked to reflect on these. Finally, a synthesis of the results of all the children were sent in an easy read format to the parent. For the children, the results were drawn in a comic strip format (see S2 Appendix).

**Phase 6: Data analysis.** Since there are no previous studies dealing with the lived experience of participation of young children, an inductive approach was used, meaning that the results were directly derived from the collected data. The data analysis consisted of five phases following Braun and Clarke (2006)'s thematic analysis guidelines.

Firstly, initial ideas were noted and shared with the first two authors. All researchers contributed to the data analysis. The first author (MC) is Occupational Therapist had extensive experience working with young children with ASD, DCD and ADHD as well as research training in child rehabilitation and is acquainted with literature on DD and participation. The second author (BD) was an Occupational therapist with experience in DD and qualitative research. All the other authors were senior researchers experienced in clinical work as well as in conducting research on DD. Our multidisciplinary team consisted of a psychologist (AD), a

**Table 1. Strategies utilized to establish rigor.**

| Strategy | Methods | Data Collection | Data Analysis | Comment |
|---|---|---|---|---|
| Credibility | Triangulation | - | √ | Three authors were involved in the analysis of the data |
| | Member Checking | √ | √ | Interview summaries were provided to all participants during the data collection. A summary of emerging themes was presented to all participants during the data analysis. |
| | Peer examination | - | √ | Three authors discussed the analysis of the data. |
| Transferability | Rich description of participants | √ | - | In the first phase of the interview, demographic information regarding the children and parents was collected. |
| Dependability | Dense description of research methods | √ | - | A detailed description of the methods is provided for study replication |
| | Triangulation | - | √ | As above |
| | | - | √ | As above |
| | | - | √ | The first and second authors recoded one week following initial coding |
| Confirmability | Triangulation | - | √ | As above |

child neurologist (AO), a rehabilitation doctor and a physiotherapist (HVW). The last author (DVDV) had extensive experience in qualitative research and participation.

The research team had no connection with the children nor their parents. Recognition of the lenses through which we analyzed the data, helped to assure that the findings reflected the children's experiences rather than our own biases or expectations. In addition, the research team adopted a realistic approach that focused on the data from the children itself, analyzing what was actually said. Consistent with this, we first generated codes from the data, then conducted thematic analysis at the semantic level, with themes being identified based upon explicit or surface meanings, rather than interpretation. Secondly, via using an iterative process, interesting data features were systematically coded. This process continued until the entire data set was coded. Analysis was done with NVivo10 qualitative data analysis Software [62]. To limit the risk of bias, the first two authors independently generated codes. Thirdly, the initial codes were discussed between first and second author who both conducted the interviews. After a mutual agreement, similarities in the codes were collated into potential themes. Via peer debriefing sessions between the first, second and last author, the researchers generated a uniform body of interwoven themes resulting in a thematic map of themes and their relationships.

In a fourth phase, the potential themes were checked by the research team to see whether the themes expressed the experiences in relation to the research question. This process resulted in a thematic map. In the fifth phase, the specificities of each theme were fine-tuned. In reality these phases were characterized by going back and forth between them, but for clarity's sake, these different phases are presented in a linear way. Table 1 provides more detailed information regarding the strategies employed to ensure data accuracy and trustworthiness.

## Results

A total of 32 middle-class parents living in the Flemish part of Belgium registered to participate in the study. Twenty-two were successfully contacted for further information. Two parents decided not to further participate after obtaining more information. Two children did not meet the inclusion criteria concerning age or intelligence quotient. Two children refused to participate in the study after discussing the informed assent with their parent.

This resulted in a total of sixteen Flemish speaking children having an average intelligence and a clinical diagnosis of DD. Two children had ASD, two had ADHD and four had DCD.

but were still in follow up for diagnostic purposes Eight children had a comorbid disorder. All children were between five and nine years old. Fifteen mothers and one grandmother joined in with the interview between the researcher and their child. As children are active participants in the research, they were invited to choose their own research name (see Table 2).

Data analysis showed that no new data emerged after conducting interviews with 11 children. The interviews of the last five children were used to check whether saturation had been reached. To be sure, we kept collecting data until successfully completing 47 individual interviews in this sample of 16 children. Only one child was able to be interviewed twice instead of three times and we could not perform a member check. In total, 3912 photographs and 68 short movies were made. It appeared that children loved the assignment as they took a lot of pictures. They enjoyed taking selfies with friends and families. The altered their pictures by adding funny things to their pictures such as hearts or frames and such. Children also took photographs of a variety of activities including pets, nature, favorite games. Before conducting the second interview, we asked the children to make a selection of the most important pictures that would help them to guide the interview. A short summary of the reoccurring pictures can be seen in S3 Appendix.

After analyzing the data, four central themes emerged (Table 3). Key quotes from children were used to illustrate each theme and to display the full range of children's perspectives.

The themes and subthemes mentioned above will be discussed at length below. In reality, these themes are related and interconnected, but for the purpose of clarity, they are discussed and presented separately.

## Theme 1: Playing

Central to this theme was that children loved to play or find play essential in their daily life. While playing, with other children, they perceived 'friendship' and 'companionship' and felt 'included'. When they played on their own in their home, the children enjoyed the sensory motor aspects of the toys. Hence, playing could be divided into two subthemes, more specifically friendship and features of activity materials.

Loeizui, a 7,5 year old boy, told us, after we asked him why person X was his friend, that he liked his friend *"because he laughs with me and because he wants to play with me and we tickle*

**Table 2. Socio-demographic characteristics and photograph information of the participants.**

| # | Chosen Name | Age | Diagnosis | Regular School | # Pictures in total | # movies | # pictures selected for the interview |
|---|---|---|---|---|---|---|---|
| 1 | Batman | 7y2m | ADHD | Yes | 58 | 2 | 7 |
| 2 | Filouke | 6y | DCD | Yes | 209 | 1 | 52 |
| 3 | Beyblade | 9y | DCD & ASD | No | 148 | 2 | 35 |
| 4 | Loeizui | 7y6m | ASD & ADHD | Yes | 457 | 12 | 61 |
| 5 | Noem | 6y6m | ADHD & DCD & ASD | Yes | 177 | 0 | 32 |
| 6 | T-Rex | 5y5m | DCD | Yes | 953 | 0 | 45 |
| 7 | Miekie | 7y7m | DCD | Yes | 248 | 37 | 27 |
| 8 | Bent | 7y4m | ASD & ADHD | Yes | 231 | 2 | 44 |
| 9 | Nardas | 6y8m | ASD & DCD | Yes | 451 | 0 | 84 |
| 10 | Mortis | 6y2m | ASD | Yes | 127 | 0 | 15 |
| 11 | Ben | 7y5m | ASD & ADHD | Yes | 79 | 0 | 7 |
| 12 | Minnie | 6y7m | DCD | Yes | 393 | 0 | 55 |
| 13 | Rudy | 7y2m | DCD & ASD | Yes | 221 | 0 | 37 |
| 14 | Lientje | 5y5m | ASD | Yes | 351 | 8 | 54 |
| 15 | Thomas | 6y5m | ASD & ADHD | No | 152 | 0 | 31 |
| 16 | Zorro | 7y | ADHD | Yes | 229 | 4 | 65 |

**Table 3. Overview of the themes as a result of the thematic analysis.**

| Main Theme | Sub-Themes |
|---|---|
| Playing together | ○ Friendship<br>○ Features of activity materials |
| Learning | ○ Appreciation, affirmation and appraisal<br>○ Environmental factors influencing learning |
| Family gatherings | ○ Socializing with friends and families leading to feelings of belonging and happiness<br>○ Conducting household chores and basic care routines as being a part of a family |

Barriers and facilitators of participation

*each other all the time"*. Hence, play was by far their most favorite activity. They mostly played with family members (siblings, nieces or nephews or grandparents), less often with children at school, or with the kids in the neighborhood. Their play took place primarily at home in the household atmosphere and less frequently in the community.

**Subtheme: Friendship.** Analyses of children's different stories showed that it was not only playing itself that made them experience feelings of happiness, satisfaction and belonging. Rather, the children valued their play when it was done with others. In addition, having someone to play with would often overrule the importance of a good match between child features and activity features. Children with DD appreciated input from others and enjoyed the appreciation they received through their own input. Asking why someone was perceived as a friend resulted unanimously in doing activities together rather than choosing someone as a friend for the sake of their character, or the way how their 'friends' behave themselves towards them. When asked what made person X their friend, it was hard for children to explain. Nardas, a boy of 6 years, simply answered: *"Because we know each other so long. We don't know each other well. But because he is my best friend, I know. We are since day care together, and we are now also in the same class."*

Not being able to participate in play with others therefore often resulted in feelings of rejection and sadness. With the help of his mother, Ben could beautifully explain that, due to his difficulties with self-regulation, sometimes the neighborhood kids did not let him play along leaving him with sad feelings of not being able to participate.

> *Ben: Because I have chased them very often and I have frightened them. And I then wanted to eat them. I just wanted to crush them. And I regretted it. That day. What I did. Then I saw what I had done. And I was ashamed of myself. So hard that I felt a bit alone. [Parent: Then you felt alone. Right honey?] Ben: [unintelligible] Then I walked back home. Alone and sad. And without friends.*

**Subtheme: Features of activity materials.** Another subtheme was that the choice of activities was led by environmental aspects, primarily the sensory motor aspects of activities. When probed as to why the children chose these toys and activities, it was hard for them to put this into words. The children preferred to show the pictures. They took a lot of pictures of their toys. Examples are Lego, Playmobil, cars, trains and dinosaurs just to name the most common photographed toys. The look and sound of the object or activity could often generate interest in children with DD. Ben elaborated greatly regarding why sword fighting was his favorite play activity.

> *Researcher: What's fun about sword fighting?*
>
> *Ben: You have to beat each other. It can sabre . . . That sound.*
>
> *[Parent: His saber gives off a sound.]*

### Theme 2: Learning

**Subtheme: Appreciation, affirmation and appraisal leading to recognition.**    Prominent within this theme was that children with DD were keen on learning and becoming more 'competent' or a better or more skilled version of themselves. They valued the appreciation, affirmation and appraisal from learning and doing/behaving the best they can. This gave them the feeling of being recognized for who they are and what they are capable of doing (e.g. feelings of competence). Noem, a 6 year old boy quoted: *"I like to go to school because I can learn. It goes easy. I made a test and I was the best of the class. I had read them all (he is in first grade and learns to read) and I did not make any mistake. And the other (he refers to the children in his class) did a few mistakes."*

As a result, children with DD valued going to school as it gave them the opportunity to learn, which they loved doing. Related to that, all children talked about features of themselves related to activities. According to them, learning was defined as being better at something (on social, motor or cognitive skills). Batman demonstrated this by saying that he loved doing sports as he got fitter from it.

The benefits of sport were *"That I can sweat a little bit. And I also have to go for a drink. Because that's good when I sweat. Then that fat is a bit out of me".* Same for Bent, 7 years old, who loves going to the boys scouts. *"I like going to the scouts, I have to make a square"* (he means formation at the start of the activity). *"I learn to do gymnastics at the scouts. I will become strong then."*

Most of the children mentioned some skills they had or even demonstrated them during the interview. They loved showing what they made during class or what they already could do (i.e. like doing a split in gymnastics). Activities were valued based on how they contributed to a specific purpose or goal of the child with DD such as learning new skills, being healthy or getting a reward after completing several activities. Salient was that children with DD preferred activities where they demonstrated their cognitive skills such as learning to write, to read and do math. Also, activities like playing soccer, basketball or learning how to swim were mentioned by almost all children with DD.

**Sub theme: Environmental factors influencing learning.**    All the children with DD mentioned some sort of support they were getting in engaging in activities. Mostly through their teacher or via material aids, but in exceptional cases also through other children. Material aids could take the form of visualizations for objects, structure of the day, feedback of their behavior, reward schedules, and so on. They could also be things that assisted or compensated a certain skill, like using a computer to write, or provide emotional support in stressful situations, such as a teddy bear or simply a pillow.

Lientje loved fiddling with her classmate's hair as it calmed her down.

When asked why she simply replied: *"That I can fiddle with her hair".*

*Researcher*: *And what do you think of that*?

*Lientje*: *That she allows it. Nice.*

*Researcher*: *And what does [classmate] think of that*?

*Lientje*: *Fine. (. . .) That she helps friends instead of being bothered by it.*

### Theme 3: Family and friend gatherings

**Subtheme: Socializing with families.**    The main observation here was that children valued 'belonging' and socializing with friends and family enormously as it had special family meanings.

It enhanced their emotional health and well-being. Celebrations at home such as holiday gatherings and birthday parties are enjoyed as well as having family or friends stay over to play or for sleepovers. Participating in these activities made children feel secure and beloved. It felt for them as 'being part' or 'belonging to'. Minnie, a 6 year old girl was vividly talking about her trip with her dad. *"Normally dad and I were going to the playground but we went to the movie. I made the choice (smiles). And (after her mom gives her a hint) before the movie we went to eat French fries. We had to wait very long"*. When asked by the researcher why she liked it, she simply replied *"It was just fun, dad and me."*

**Subtheme: Household chores and basic care routines.** Children all mentioned basic care routines when discussing their participation. More specifically, brushing their teeth was often something that required a lot of 'issues' such as not liking the taste of toothpaste.

Regarding bed time routines, children valued enormously the help provided to facilitate their participation by reading a story before going to bed or having a little light on. Children mentioned primarily the help received from their mothers. On some occasions, they mentioned the aid from their fathers.

Children with DD also valued the appreciation when helping their mother by performing household chores such as taking care of a pet, unloading groceries or scooping and pouring or setting the table. Filouke, a 6 year old boy, said *"I love getting up in the morning as the first. Then I can feed the fish and the cat"*.

**Theme 4: Barriers and facilitators of participation.** The data revealed that children with DD spoke unanimously in a positive way about their current everyday life. It seemed that they did not compare themselves with what other children can or cannot do. If there was a desire for change, it was because feelings of inclusion, recognition and belonging were lacking. If that was the case, they used two strategies to deal with it.

Or children with DD just accepted it as it is and complied and decided not to participate. This was particularly the case if others didn't make remarks about them not participating.

*Rudy, 7 years of age and going to a school specialized in children with autism, explained that he wanted to go back to his previous school as at that school he wasn't forced to learn to play with other children.* When asked why he wanted to go back to the mainstream educational system, he commented that he–due to his autism, had a hard time making friends and at the special school, he was forced to learn this.

The other way of reacting to it was by asking for help, primarily to the mother/teacher to enable their participation. The latter way was conducted when children with DD perceived the benefit of 'connection' with others while participating.

It seemed that the need for joining in participation and doing their best seemed to be driven by 1) the need of 'belonging' or being with others and 2) being valued by others for who they are and what they are 'competent' or capable of doing. Nevertheless, they were aware that they sometimes weren't able to participate due to the mismatch between their capacity and requirements/skills needed to be able to participate. Only then, children with DD in this study asked for help. Strikingly, mothers were perceived by the children as gatekeepers and facilitators of their participation at home and in the community. Teachers were considered to be the participation facilitators at school. Beyblade phrased this marvelously.

*Beyblade*: *I have not made a mistake yet. But I have a secret weapon for that. (. . .) My mom.*

*Researcher*: *Your mom is your secret weapon. That's nice. And what makes her your secret weapon?*

*Beyblade*: *If I do not know something, I ask her.*

The most obstructing barriers perceived by the children with DD were primarily situated at the level of a 'mismatch' between the skills or 'competences' children possess and the requirements needed to participate within a specific activity. Some children mentioned more child related barriers such as not being able to meet the physical demands (e.g., strength, endurance, coordination), cognitive demands (e.g., concentration, attention, problem-solving) and social demands (e.g., communication, interacting with others) of typical activities. When asked what Filouke, 6 years old, wanted to change regarding his participation, he articulated this nicely by telling that he did not want to fall anymore.

*Researcher*: Do you fall often?

*Filouke*: Yes.

*Researcher*: Can you give an example?

*Filouke*: I have stitches in my chin. And here too. \*Indicates eyebrow.\*

*Researcher*: Oops. How did that happen?

*Filouke*: I was on number one on my gears. And actually, that is for the mountain. And I had fallen because that was too fast.

*Researcher*: So you fall when you cycle.

All children with DD perceived their mothers and grandparents as enabling their participation. By offering emotional support, assisting in planning activities, helping to make choices (important for the feeling of 'autonomy'), solving problems and to perform a task or taking on parts of an activity where the child still lacked the needed skills are all perceived by the children as enabling their participation.

Children often mentioned gatekeepers to access activities. Other people provided access to activities or made it completely impossible. Again, the biggest influence was found for mothers who allowed certain activities or certain friends or imposed rules and conditions for participation in certain activities. Mothers also provided access to activities by making arrangements.

*Researcher*: Ben. Can you do anything that you want to do?

*Ben*: No. I can't watch the TV station Kadet (where only cartoons on action figures and such is played). *That is prohibited. Not anymore.*

[*Mother*: Why has mommy banned Kadet?]

*Ben*: That's not good for me.

[*Mother*: That makes you very restless.]

## Thematic map

By grouping related themes together, visualizing connections with arrows and mediators as gates, the data can be presented as follows. Fig 2 represents the subjective aspects of participation of young children with DD. For successful participation to result, there must be a good match between child's capacities and the characteristics of the activity the child wants to do.

There is a positive feeling if 1) the child experiences friendship and feels included through playing with friends, 2) feels appreciated and receives appraisal and approval leading to recognition for who they are by learning and 3) feels a sense of belonging to a group through joining

family gatherings. Context factors, materials, parents and peers or friends are represented in the figure as mediators to access activities and therefore participation. When there are—on occasions—moments that their participation was obstructed, the children used two strategies to resolve the problem. Or they walked away from it and choose not to participate, or when autonomously motivated for the activity, they relied primarily on their context (i.e. mothers) as enabling their participation.

## Discussion

The principal goal of this Photo Elicitation study was to investigate how children with DD define their participation.

The results not only gave us an insight in the fact that children with DD perceived their participation as satisfying when they can play, learn and join in family gatherings.

Put differently, when they sense feelings of inclusion, recognition (or competence) and belonging. The child's experience of playing with friends, learning at school and doing things together with the family, matched with the existing evidence of the construct participation. Themes such as self-awareness, identity, confidence, preference, motivation, interests, satisfaction and self-esteem are proposed as part of the participation construct [5, 15, 35, 63].

This research also revealed that participation did not just happen within a frame of contextual or environmental factors, but that those factors served, together with the parents and peers, as gatekeepers or mediators to the participation of children with DD in activities. This indicates that for children, participation should be regarded as an interrelationship of constructs such as activity, body function components and environmental factors of the ICF.

While some of the findings of this study are not surprising given recent systematic review of the literature, to our knowledge this is the first study to specifically ask young children with DD about their own subjective experience of participation. The quotations from the young children highlight that, differently than for the older children, they are surprisingly satisfied with their participation.

This may suggest that little obstruction is perceived by the children thanks to the vast support of primarily their mothers and other contextual or environmental factors.

This finding can be related to or interpreted along others theories such as 1) the Self-Determination Theory (SDT)[64], 2) the theory regarding Belonging, 3) the theory on Recognition and 4) on the research on Happiness that are not commonly used nor known in the rehabilitation guiding theories and frameworks. It would be interesting to conduct further research on the use of or the relationship with these theories to better guide clinical pediatric rehabilitation.

First, the theory regarding the SDT, more specifically the feeling of autonomy, belonging and competence seems important to include when developing interventions as health care clinicians. The findings of this study indicate that the benefits of SDT are not only interesting for the engagement and well-being of regular students. This study seems to indicate the value of including the principles of SDT for children with DD as well.

Second, a same argument can be made regarding the theory of Belonging. According to [65] "belonging" has been described as a puzzling term. Feeling a sense of belonging (or not), being morally, socially or legally recognized as belonging (or not), could have the power to change lives. As the literature on belonging is currently very diverse, it might be interesting to further investigate how this construct relates to participation and young children with DD.

Third, according to [66], "recognition" is a key concept in contemporary social and political thought. The fundamental insight associated with this concept is that we cannot really be who we are or who we want to be if others do not treat us in certain ways. Leopold (2018) argues

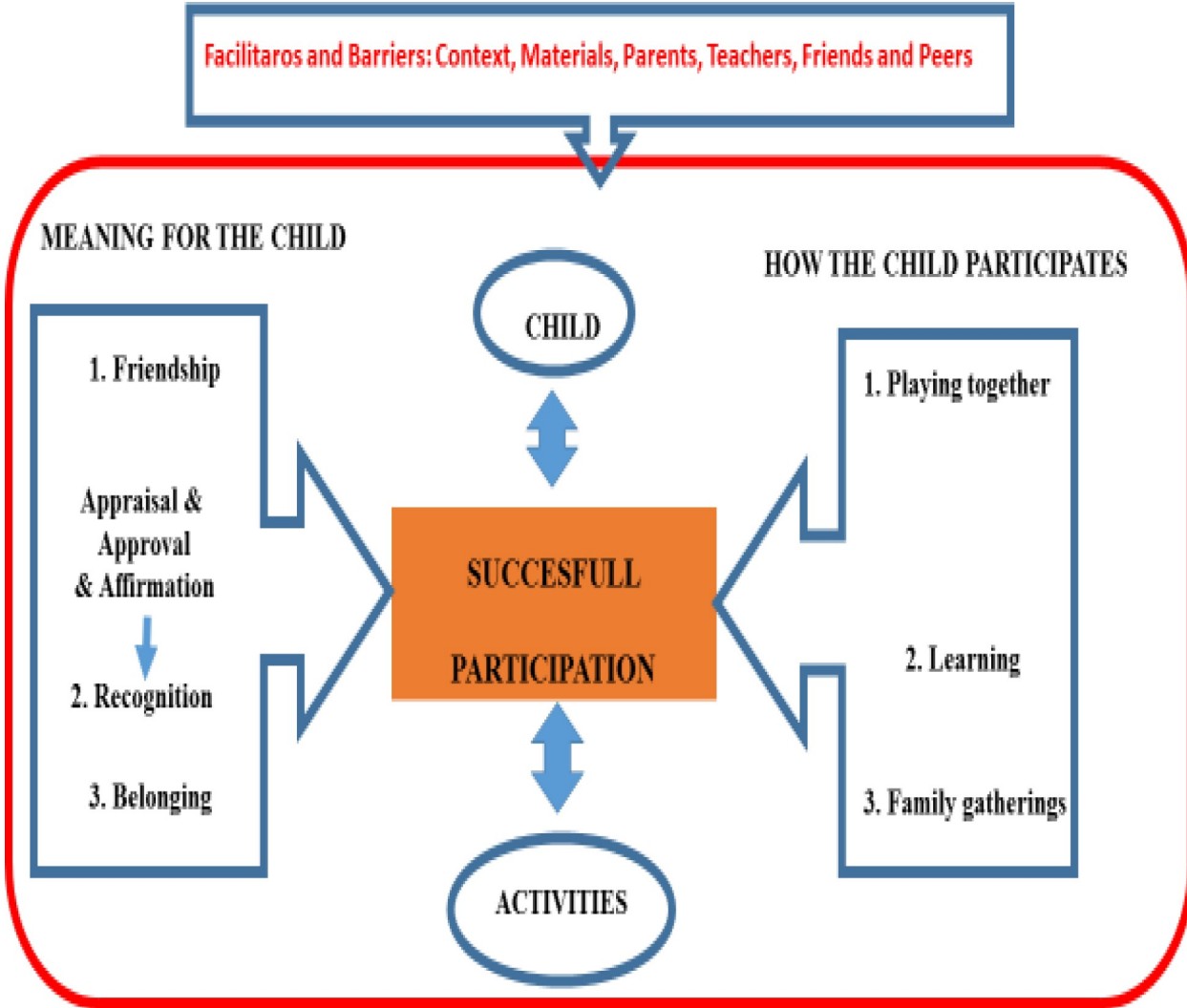

**Fig 2. Thematic map of the findings.**

that we depend on others or, more specifically on the recognition by others, in order to be able to actualize our identities. She argues that if we do not find recognition in our personal interactions or in the wider society, we are peculiarly constrained in our being.

Fourth, it seems worthwhile to further investigate whether the construct of happiness and participation could be related. The term happiness is used in the context of mental or emotional states, including positive or pleasant emotions ranging from contentment to intense joy. According to [67] traditional mental health models focus on psychological problems and distress. The recent emergence of positive psychology has placed increased emphasis on the positive factors, such as life satisfaction and happiness [67, 68]. Given the importance of positive indicators of emotional well-being, particularly acknowledging that high happiness contributes to determining remission of psychological symptoms, the relationship between happiness and participation should be further investigated. Given the fact that young children with DD have often adaptive-behavior deficits, it seems worthwhile to further investigate the clinical use of indices of happiness offers. Quite simply, the absence of indices of happiness or presence of indices of unhappiness can cause occasional changes in programming. By observing affective

behavior, school, leisure, and therapeutic activities, can be added to or subtracted from the child's support plan to ensure more or better participation possibilities.

To conclude, looking further into detail in how these theories can enhance our thinking seems important as it advances understanding by identifying specific factors (i.e. playing, learning and family gatherings) that decision-makers and interventionists may consider to help children to feel that they belong, are recognized, happy and therefore be better equipped to participate.

Since the interviews focused on the child, little data was gathered concerning the perception of the parent. Nevertheless, our data revealed that parents were more aware of obstacles than the children themselves. Similar results are in line with the findings of Jasmin, Tetreault, Lariviere and Joly [69], proving that, both perceptions might differ in that aspect. Further comparative research on both perceptions might confirm or disprove this suggestion.

It also could be that, as children grow older, the gap between the child's capacity and requirements to be able to participate enlargers. It could be that mothers are afraid to be less able anymore to facilitate their child's participation resulting in a decreased sense of participation in older children with DD. This however, needs further longitudinal research to untangle this idea.

In the meantime, if parents experience more needs concerning the participation of their child than the children themselves, more parent training and coaching interventions are advised to mitigate this discrepancy. More specifically, data proposes that parent coaching in matching activity factors with child factors and removing contextual obstacles to participation would result in more access to activities [15, 63, 70, 71]. Finally, our results suggest that health care workers and specialists in early childhood education should extricate participation from development so that the challenges of healthy growth and development are met.

Focusing on development suggests that there is a typical or right way of performing activities, rather than principles related to the effectiveness of the activity outcome (e.g. that the child is engaged with, or is undertaking, the desired task and completing it in its own way [5]. This change in focus of providing health care to young children might better help young children with DD to reach their full potential throughout their lifespan.

To conclude, as children experienced obstruction in their participation due to body functions (e.g. over-sensitivity to noise or being unable to regulate your emotions while playing), interventions focusing on tackling this seem advisable. As parents and peers appear to act as gatekeepers towards the participation of children with DD, interventions focusing on these contextual factors might also be needed [72].

## Limitations

All studies have their limitations. In this study one of the limitation was that we had to recruit children through their parents. For several reasons some parents refused to allow their child to participate. Consequently, there might be some bias in our results since we only heard the voices on consenting participants. Moreover, although we tried to obtain a maximum variation sample, few participants belonged to lower socio-economic families.

Another possible limitation is that, according to Docherty & Sandelowski [73], children of all ages might withhold emotion-laden information. It seems that children tend to mask negative feelings or disappointment because they do not want to elicit a negative response from the interviewer or others who might hear what they say. To overcome this, we made sure to avoid the display of feelings such as surprise or horror at what the child was reporting. Children were also reassured that their personal stories would not be revealed to the people (especially parents) they are trying to protect them from these negative feelings. It might have been that

by not including parents in the room where the interview took place, that the children would have reported more negative feelings than they currently did. Additional studies seems indicated with more children with DD living in lower socio-economic families.

In addition to the already aforementioned limitations, this study was carried out on a small sample of children with DD in Flanders (Dutch-speaking Belgium). Studies with a larger sample, and perhaps also other age groups, cultural backgrounds and other DD seem advisable. As well, we only focused on a time period of 'one week' rather than following these children during a longer period in time. More time in the field to gather longitudinal and observational data would capture participation across the different seasons or times of the year. As well, gathering data from other groups of participants such as those with differences in urban-rural, gender, ethnicity is still necessary.

## Conclusion

Because the main goal of this study was to define how children with DD experience participation, it was a challenge to find a way to assess participation in these young children. Our study revealed that it was possible by adapting the informed consent and by using open interviews with the child and parent, supported by a camera assignment. This study generated rich data on the subject, confirming that young children with DD were able to inform us accurately on their participation experiences. Therefore, researchers have to treat accounts of children of their own experiences as valid in their own right.

Looking more closely at their descriptions, we noticed that they perceive their participation as satisfying when they can play, learn and join in family gatherings. In other words, when they sense feelings of inclusion, recognition (or competence) and belonging. Related to the data, children discussed themes related to their person, activities, connections and mediators between those themes. These themes fit well within earlier and current research on the subject of *participation*

To conclude, the results of the study contribute to a deeper understanding of the subjective experience of participation of young children with DD, and thereby add to the construct validity of the concept of participation.

## Supporting information

**S1 Appendix. Interview protocol.**
(DOCX)

**S2 Appendix. Dissimination to the children.**
(PDF)

**S3 Appendix. Short summary of the photographs.**
(DOCX)

## Acknowledgments

The authors would like to thank the children and their parents who participated so willingly in this project.

## Author Contributions

**Conceptualization:** Marieke Coussens, Birger Destoop, Dominique Van de Velde.

**Data curation:** Marieke Coussens, Birger Destoop.

**Formal analysis:** Marieke Coussens, Birger Destoop, Dominique Van de Velde.

**Funding acquisition:** Marieke Coussens, Birger Destoop.

**Investigation:** Marieke Coussens, Birger Destoop.

**Methodology:** Marieke Coussens, Annemie Desoete, Dominique Van de Velde.

**Project administration:** Guy Vanderstraeten, Dominique Van de Velde.

**Resources:** Marieke Coussens.

**Software:** Marieke Coussens, Birger Destoop.

**Supervision:** Annemie Desoete, Guy Vanderstraeten, Hilde Van Waelvelde, Dominique Van de Velde.

**Validation:** Marieke Coussens.

**Visualization:** Marieke Coussens.

**Writing – original draft:** Marieke Coussens, Birger Destoop.

**Writing – review & editing:** Birger Destoop, Stijn De Baets, Annemie Desoete, Ann Oostra, Hilde Van Waelvelde.

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
