## [Decision Letter · Decision Letter 0]

11 Nov 2019

PONE-D-19-22303

A qualitative photo-voice action research study to elicit the perception of young children with Developmental Disabilities such as ADHD and/or DCD and/or ASD on their participation.

PLOS ONE

Dear Mrs Coussens,

Thank you for submitting your manuscript to PLOS ONE. After careful consideration, we feel that it has merit but does not fully meet PLOS ONE’s publication criteria as it currently stands. Therefore, we invite you to submit a revised version of the manuscript that addresses the points raised during the review process.

Although the Reviewers appreciated the Authors' effort to fill a gap in qualitative research related to children with Developmental Disabilities, they have nevertheless highlighted important limitations of the study that require a careful and thorough revision, especially in its methodological aspects. I therefore recommend to the Authors to address all the Reviewers’ comments to make the manuscript suitable for publication.

We would appreciate receiving your revised manuscript by Dec 26 2019 11:59PM. To enhance the reproducibility of your results, we recommend that if applicable you deposit your laboratory protocols in protocols.io, where a protocol can be assigned its own identifier (DOI) such that it can be cited independently in the future. For instructions see: http://journals.plos.org/plosone/s/submission-guidelines#loc-laboratory-protocols

We look forward to receiving your revised manuscript.

Kind regards,

Stefano Federici, Ph.D.

Academic Editor

PLOS ONE

Journal Requirements:

1. We noticed you have some minor occurrence of overlapping text with the following previous publication(s), which needs to be addressed:

- Maciver, Donald, et al. "Participation of children with disabilities in school: A realist systematic review of psychosocial and environmental factors." PloS one 14.1 (2019): e0210511.

- Antshel, Kevin M., and Natalie Russo. "Autism spectrum disorders and ADHD: Overlapping phenomenology, diagnostic issues, and treatment considerations." Current psychiatry reports 21.5 (2019): 34.

- Zwicker, Jill G., et al. "Developmental coordination disorder is more than a motor problem: children describe the impact of daily struggles on their quality of life." British journal of occupational therapy 81.2 (2018): 65-73.

 The text that needs to be addressed involves some sentences of the introduction.

In your revision ensure you cite all your sources (including your own works), and quote or rephrase any duplicated text outside the methods section. Further consideration is dependent on these concerns being addressed.

2. Please include a caption for figure 2

Additional Editor Comments (if provided):

Although the Reviewers appreciated the Authors' effort to fill a gap in qualitative research related to children with Developmental Disabilities, they have nevertheless highlighted important limitations of the study that require a careful and thorough revision, especially in its methodological aspects. I therefore recommend to the Authors to address all the Reviewers’ comments to make the manuscript suitable for publication.

Reviewers' comments:

Reviewer's Responses to Questions

**Comments to the Author**

1. Is the manuscript technically sound, and do the data support the conclusions?

Reviewer #1: Partly

Reviewer #2: Yes

2. Has the statistical analysis been performed appropriately and rigorously? 

Reviewer #1: N/A

Reviewer #2: N/A

3. Have the authors made all data underlying the findings in their manuscript fully available?

Reviewer #1: Yes

Reviewer #2: Yes

4. Is the manuscript presented in an intelligible fashion and written in standard English?

Reviewer #1: Yes

Reviewer #2: Yes

5. Review Comments to the Author

Reviewer #1: This is a timely and interesting paper that aims to address a lack of qualitative data on the experiences and perceptions of young children on their participation. However, there are a number of areas that warrant more attention to strengthen the report and augment the reporting of the study.

Research design: the research design is described as an action research study which is confusing to the reviewer as it was not defined clearly and there are many diverse ways to consider action research. Usually however, it involves an equal attention to taking part in making some change or doing some advocacy as part of the design which is not the case in this study. This study includes the use of photographs which can be considered participatory for sure so perhaps this is where the confusion lies. Clarification on the description of the design and choice of methods would be important.

Inclusion and exclusion criteria: in page 8, the inclusion criteria lists the diagnostic requirements of each child needing to have ' at least one of the 3 diagnoses,' then further down on line 170 it states: 'children were excluded when there was only a presumption of one of the 3 diagnoses'. Yet on page 13, line 268 there is a sentence that states: 'eight children had a comorbid disorder. This appears to imply that the remaining 8 had only 1 of the disorders, but this was an exclusion condition. Please can you clarify across the paper for consistency, what exactly the inclusion and exclusion criteria was, and to clarify consequently the list of 16 children and their profiles on table 2.

Reporting the findings:

Analysis of the photos is lacking in detail and the range and breadth of diverse activities is not documented. In other studies children taking photographs usually include varied activities that can include pets, nature, favorite games etc and are not only social activities which is what this study reports: the first theme listed is playing together. This therefore presents a theme that may not reflect the full picture of participation, which is often in the physical environment as well as the social. Further detail of the synthesis of all the photos children took would strengthen the paper as it would provide a more robust profile of the range of activities the children documented in their task.

Themes seem to be presented differently with some themes being larger than others, with some sub themes such as activity materials hardly consisting of a paragraph which is problematic as it does not present enough data to convince robustness. Some themes appear to be unresolved-for example under the theme of learning there is some data about sports and participation that does not clearly relate to learning- more explanation of why it is analyzed and situated underneath this theme is needed. Furthermore, the sub theme on environmental factors is unclear in its present form and needs more explanation. Overall the themes need to reflect multiple dimensions to be considered robust-to include the negative as well as positive sides for examples, or to include opposite opinions among the children's data as well as things they agreed with.

Finally the discussion would need more focused work once the findings are more clearly described. There are afew assumptions being made in the discussion. For example, it is important to note that children with DD have been involved in other studies similarly about their engagement in activities and their experiences, so the authors need to say that this is the first study to their knowledge, (line 514). Also, an assumption is made that mothers aren't able to enable participation as the child gets older (line 533), this needs to be reconsidered.

Consent- as all studies require consent, this is not the most useful or insightful limitation to make in the final limitation section. Reconsider including other limitations such as more time in the field to gather longitudinal data which would therefore capture participation across the different seasons or times of the year, or to gather data from other groups such as those with differences in urban-rural, gender, ethnicity.

Although the paper is well referenced, consider looking at more of the literature on happiness as an indicator of well being in young children to inform a more nuanced data analysis outcome and discussion. This aspect appeared to be evident in your data but not considered from this perspective.

Reviewer #2: Dear authors

it was a pleasure to learn from your work. We would suggest following elements to bring the article on 'the next level'.

1. on page 4, line 79 you were bringing research into ' ' : I would suggest to skip the ' ' as it can be seen as an act of non-respect to your peers

2. at page 6 you conclude that the opinion of children is not yet studied and you are calling this ...a suprise...: are there any reasons why this voice is not put forward? And is the idea of children having 'their own voice' not a bit naive I think that more can be said to make sure the surprise gets context

3. photo voice projects arre more and more using personal phones as a tool instead of digital camera's: did you think about this option?

4. how did you prepare the children about the ethical standards necessary to work with photo voice??

5. if children felt the comic strip style of report was a bit childish , could they get an alternative version?

6. what was the rationale to use Thematic Analysis as analytic tool?

7. on page 11 your are offering a good overview about the members of your research team; we did not get information about 'the position' of this team visà vis the children involved in your study

8. is it necessary/helpful to bring in the diagnosis of the children linked to theier quotes (this sound very medical model)

9. was it an option to bring in certain pictures in the text to make your report more vivid?

10. technical: you named figure 2, figure 1

11. why do you skip the 'survival strategies of the kids' (walking away,...) in your figure 2? Now it sticks to the succesfull participation. We all know that 'failures' can help to learn, and that a lot of the kids know failures very well..

12. I think that in your conclusions the Self Determination Theory gets to much attention. I can agree with the fact that researchers are looking for frameworks that help to bring elements together, but... now you are ignoring great literature about 'recognition' (Judith Butler e.g.) and their is an enormous amoutn of interesting literature about 'belonging'

13. on page 26 you are observing the fact that servie providers are not in the stories and you are bringing this back to the culture of Flemish support giving services. I can't follow this linear way of thinking. I think that younger children always stick to the concrete elements of a procedure (most interviews were done at home with parents) - a lot of children are living in middle claas families (with a lot of time and car energy). I would suggest to look in that direction

warm regards

Geert Van Hove, Prof.dr.

6. PLOS authors have the option to publish the peer review history of their article (what does this mean?). If published, this will include your full peer review and any attached files.

Reviewer #1: No

Reviewer #2: Yes: Geert Van Hove, Prof.dr.

---

## [Author Response · Author response to Decision Letter 0]

3 Jan 2020

Point by point answer to the reviewers.

Dear Editor, Dear reviewers,

Thank you for this additional feedback. We have tried to make all necessary changes and a point to point response can be found in the section where all the documents are uploaded.

We hope this responds to the remarks, but we are willing to make any changes if this is not the case. We also uploaded our point by point answer to the editor in the section where all the documents and main manuscripts and figures and tables are uploaded. 

Sincerely,

Marieke Coussens, on behalf of the entire research team.

---

## [Decision Letter · Decision Letter 1]

10 Feb 2020

A qualitative photo-elicitation research study to elicit the perception of young children with Developmental Disabilities such as ADHD and/or DCD and/or ASD on their participation.

PONE-D-19-22303R1

Dear Dr. Coussens,

We are pleased to inform you that your manuscript has been judged scientifically suitable for publication and will be formally accepted for publication once it complies with all outstanding technical requirements.

With kind regards,

Stefano Federici, Ph.D.

Academic Editor

PLOS ONE

Additional Editor Comments (optional):

Reviewers' comments:

Reviewer's Responses to Questions

**Comments to the Author**

1. If the authors have adequately addressed your comments raised in a previous round of review and you feel that this manuscript is now acceptable for publication, you may indicate that here to bypass the “Comments to the Author” section, enter your conflict of interest statement in the “Confidential to Editor” section, and submit your "Accept" recommendation.

Reviewer #2: All comments have been addressed

2. Is the manuscript technically sound, and do the data support the conclusions?

Reviewer #2: (No Response)

3. Has the statistical analysis been performed appropriately and rigorously? 

Reviewer #2: (No Response)

4. Have the authors made all data underlying the findings in their manuscript fully available?

Reviewer #2: (No Response)

5. Is the manuscript presented in an intelligible fashion and written in standard English?

Reviewer #2: (No Response)

6. Review Comments to the Author

Reviewer #2: (No Response)

7. PLOS authors have the option to publish the peer review history of their article (what does this mean?). If published, this will include your full peer review and any attached files.

Reviewer #2: Yes: Geert Van Hove

---

## [Editor Report · Acceptance letter]

20 Feb 2020

PONE-D-19-22303R1 

A Qualitative Photo Elicitation Research Study to elicit the perception of young children with Developmental Disabilities such as ADHD and/or DCD and/or ASD on their participation. 

Dear Dr. Coussens:

I am pleased to inform you that your manuscript has been deemed suitable for publication in PLOS ONE. Congratulations! Your manuscript is now with our production department. 

With kind regards,

on behalf of

Prof. Stefano Federici 

Academic Editor

PLOS ONE